# Development and Evaluation of a Quantitative Systems Pharmacology Model for Mechanism Interpretation and Efficacy Prediction of Atezolizumab in Combination with Carboplatin and Nab-Paclitaxel in Patients with Non-Small-Cell Lung Cancer

**DOI:** 10.3390/ph17020238

**Published:** 2024-02-12

**Authors:** Chen-Yu Wang, Hao-Ran Dai, Yu-Ping Tan, Di-Hong Yang, Xiao-Min Niu, Lu Han, Wen Wang, Ling-Ling Ma, Aleksi Julku, Zheng Jiao

**Affiliations:** 1Department of Pharmacy, Shanghai Chest Hospital, Shanghai Jiao Tong University School of Medicine, Shanghai 200030, China; wangchenyu@shchest.org (C.-Y.W.); daihaoran1212@foxmail.com (H.-R.D.); phoebetan1994@126.com (Y.-P.T.); yangdh@zjcc.org.cn (D.-H.Y.); uger123@sjtu.edu.cn (L.H.); 2Department of Pharmacy, Zhejiang Cancer Hospital, Institute of Basic Medicine and Cancer (IBMC), Chinese Academy of Sciences, Hangzhou 310022, China; 3Department of Oncology, Shanghai Chest Hospital, Shanghai Jiao Tong University School of Medicine, Shanghai 200030, China; ar_tey@hotmail.com; 4Puissan Biotech Oy, 00510 Helsinki, Finland; wen.wang@puissan.com (W.W.); lingling.ma@puissan.com (L.-L.M.);

**Keywords:** quantitative systems pharmacology, atezolizumab, nab-paclitaxel, carboplatin, non-small-cell lung cancer

## Abstract

Immunotherapy has shown clinical benefit in patients with non-small-cell lung cancer (NSCLC). Due to the limited response of monotherapy, combining immune checkpoint inhibitors (ICIs) and chemotherapy is considered a treatment option for advanced NSCLC. However, the mechanism of combined therapy and the potential patient population that could benefit from combined therapy remain undetermined. Here, we developed an NSCLC model based on the published quantitative systems pharmacology (QSP)-immuno-oncology platform by making necessary adjustments. After calibration and validation, the established QSP model could adequately characterise the biological mechanisms of action of the triple combination of atezolizumab, nab-paclitaxel, and carboplatin in patients with NSCLC, and identify predictive biomarkers for precision dosing. The established model could efficiently characterise the objective response rate and duration of response of the IMpower131 trial, reproducing the efficacy of alternative dosing. Furthermore, CD8+ and CD4+ T cell densities in tumours were found to be significantly related to the response status. This significant extension of the QSP model not only broadens its applicability but also more accurately reflects real-world clinical settings. Importantly, it positions the model as a critical foundation for model-informed drug development and the customisation of treatment plans, especially in the context of combining single-agent ICIs with platinum-doublet chemotherapy.

## 1. Introduction

Despite advancement in cancer research and treatment, lung cancer continues to be the primary cause of cancer-related fatalities globally, with 1.8 million new deaths in the year 2020 [1]. Approximately 85% of all lung cancer cases are classified as non-small-cell lung cancer (NSCLC) by histological subtypes [2]. Although chemotherapy has been proven to be effective, the overall response and survival rates in NSCLC remain low among patients without driver mutations [3].

The emergence of immune checkpoint inhibitors (ICIs) targeting cytotoxic T-lymphocyte antigen-4 (CTLA-4), programmed death 1 (PD-1), and programmed death ligand 1 (PD-L1) has heralded novel approaches toward the treatment of NSCLC [4]. ICI monotherapy or combined with chemotherapy demonstrated favourable benefits in NSCLC patients without driver mutations; however, the underlying mechanism was not quantitatively investigated [3].

Quantitative systems pharmacology (QSP) is a mechanistic modelling approach used to assess therapeutic interventions for diseases by linking the molecular and cellular mechanisms of diseases and drugs to system-wide dynamics, bridging biomarkers, and disease-relevant clinical endpoints [5]. Recently, a QSP model platform for immuno-oncology (IO) that incorporates detailed mechanisms for important immune interactions was developed, allowing for the construction of QSP models of IO with varying degrees of complexity [6]. Based on the platform, several QSP models were developed to identify predictive biomarkers and predict efficacy for various ICIs [7,8,9,10,11]. However, these models are only limited to single ICI or dual combinations and are incapable of describing triple-drug combinations.

Recently, several first-line phase 3 trials investigated and confirmed the efficacy and safety of a combination of single agent of ICI with platinum-doublet chemotherapy in patients with advanced NSCLC [12,13,14]; however, the underlying biological mechanisms remain unexplored. Therefore, in this study, we developed and validated a QSP model for NSCLC using the published IO platform based on the data from the IMpower131 study (NCT02367794) [12] to explore the mechanisms of triple combined therapy and identify accessible biomarkers for predicting the clinical outcomes for atezolizumab plus carboplatin and nab-paclitaxel combination therapy.

## 2. Results

### 2.1. Model Development

#### 2.1.1. Model Structure

The primary compartments and modules of the QSP model were constructed based on prior triple-negative breast cancer (TNBC) studies. Within the QSP model, the dynamics of cancer cell growth and death was described by the cancer module. The antigen module described the release of both self- and cancer-associated antigens from dead cancer cells, whereas the antigen-presenting cells (APC) module described the antigens captured by mature antigen-presenting cells (mAPC) migrating via the lymphatic vessels to the tumour-draining lymph nodes (TdLNs). The effector T cells (Teff) and regulatory T cells (Treg) module described the mAPC-mediated activation of both naïve CD8+ and CD4+ T-cells, converting them into Teffs and Tregs, respectively. Following activation, Teff and Treg cells are known to leave the lymph nodes, circulate through the bloodstream, and, upon extravasation, localise to the tumour and other peripheral tissues. Within the tumour microenvironment, Teff cells engage in the direct elimination of cancer cells but can become exhausted due to suppressive actions by Tregs and the PD-1/PD-L1 interaction with tumour cells. However, atezolizumab enhances the anti-tumour response by preventing the PD-1/PD-L1 interaction, thereby facilitating the unrestricted activity of Teff cells against tumour cells, as described by the checkpoint module. Additionally, the nab-paclitaxel and carboplatin modules described the targeting of cancer cells via the cytotoxic effects of nab-paclitaxel and carboplatin. The established QSP model comprised 216 ordinary differential and 55 algebraic equations. These model equations are presented in Appendix A.

This QSP model mainly extends the equation of the time-varying number of cancer cells in a tumour, which is expressed as follows:(1)$$\begin{array}{cc} \frac{dC}{dt}=k_{C,growth} & \times C\times log\left(\frac{C_{total}}{C_{max}}\right)\\ & -[k_{C,death}+k_{C,nabp}\times \frac{NabP}{NabP+{IC50}_{nabp}}\times f_{c,nabp}\\ & +k_{C,carb}\times \frac{carb}{carb+{IC50}_{carb}}\times f_{c,carb}\\ & +k_{C,T}\times \frac{T}{T+K_{C,T}C_{total}}\times \frac{T}{T+K_{C,Treg}T_{reg}}\times \left(1-H_{PD1}\right)\\ & \times \left(1-H_{TGF}\right)\times \left(1-H_{MDSC}\right)]\times C \end{array}$$
where C is the count of cancer cells; $k_{C,growth}$ is the cancer cell growth rate; $C_{total}$ is the total count of cancer cells in the tumour; $C_{max}$ is the maximal capacity of cancer cells in the tumour; $k_{C,death}$ is apoptosis caused by natural cell death or natural killer cells; $k_{C,nabp}$ is the cytotoxic activity by nab-paclitaxel incorporated with a rate constant; NabP is the concentration of nab-paclitaxel in tumour; ${IC50}_{nabp}$ is the half-maximal nab-paclitaxel concentration for cancer cell killing; $f_{c,nabp}$ is the fraction of cancer cells that is accessible by nab-paclitaxel; $k_{C,carb}$ is the cytotoxic activity by carboplatin incorporated with a rate constant; carb is the concentration of carboplatin in the tumour; ${IC50}_{carb}$ is the half-maximal carboplatin concentration for cancer cell killing; $f_{c,carb}$ is the fraction of cancer cells that is accessible by carboplatin; $k_{C,T}$ is the maximal killing rate of Teff; T is the count of Teff cells; $T_{reg}$ is the count of Treg cells; $K_{C,T}$ and $K_{C,Treg}$ represent the inhibitory effects of cancer cells and Tregs, respectively; and $H_{PD1}$, $H_{TGF}$, and $H_{MDSC}$ are the inhibitory effects of PD-1, TGF-β, and myeloid-derived suppressor cells (MDSCs), respectively.

We also optimised the rule of cancer cell capacity using tumour diameter and spherical calculation formulas based on a previous study on NSCLC [6].
(2)$$C_{max}=\frac{3}{4}\times \pi \times {\left(\frac{{DIA}_{T,\;max}}{2}\right)}^{3}\times {DEN}_{T,cell}$$
where $C_{max}$, ${DIA}_{T,max},$ and ${DEN}_{T,cell}$ represent the maximal cancer capacity, maximum tumour diameter, and cancer cell density, respectively.

#### 2.1.2. Model Parameters

The established QSP model comprised 277 parameters. The lists of model parameters, reactions, algebraic equations, and cellular and molecular species are presented in Appendix A. Notably, 14 of the 277 parameters were specifically dedicated to the growth and death of cancer cells and were informed by clinical and experimental evidence derived from NSCLC clinical studies. We also identified 26 parameters that required the estimation of their distributions to represent inter-individual variability.

Parameter sensitivity analysis (PSA) analysis showed that the tumour-specific T-cell clone, maximum clearance rate of nab-paclitaxel from the central compartment, and rate of cancer cell death by Teff cells were the most important parameters correlated with the percentage decrease in tumour volume (Figure 1B). Conversely, we observed that the cancer cell growth rate, half-maximum concentration of nab-paclitaxel required for cancer cell eradication, and rate of angiogenic factor induction by nab-paclitaxel were the most critical parameters correlated with the percentage increase in tumour volume (Figure 1). The distributions of selected parameters are presented in Appendix A.

### 2.2. Model Evaluation

In the carboplatin plus nab-paclitaxel arm, the model predicted an objective response rate (ORR) of 40.5% and duration of response (DOR) of 5.3 months. For the atezolizumab combined with carboplatin and nab-paclitaxel arm, the predicted ORR and DOR were 52.0% and 7.1 months, respectively. These predictions showed biases under 5% and were consistent with findings from the IMpower131 study (Table 1). 

As shown in Figure 2, the time-varying responses and waterfall plots displaying tumour dynamics were consistent with those observed in the IMpower131 study, implying the reliability of our established QSP model.

Moreover, three atezolizumab dosage regimens [840 mg every 2 weeks (Q2W), 1200 mg every 3 weeks (Q3W), and 1680 mg every 4 weeks (Q4W)] combined with the same chemotherapy regimens (carboplatin at an area under the concentration–time curve (AUC) of 6 mg/mL/min intravenously (IV) on day 1 and nab-paclitaxel at 100 mg/m^2^ IV on days 1, 8, and 15) led to similar changes in tumour volume and tumour diameter (Table 2). These results further demonstrated the favourable predictive performance of the model.

### 2.3. Model Application

The distributions of potential predictive biomarkers for both response and non-response subgroups are shown in Figure 3. We found that PD-L1 expression was a beneficial biomarker for the triple-drug therapy, which agrees with the results of the subgroup analysis of the IMpower131 study [12].

We also identified that the CD8+ and CD4+ T cell counts and Treg cell densities in tumours were significantly higher in responders than in non-responders, which is consistent with the findings of previous studies on NSCLC [7,9]. In the receiver operating characteristic (ROC) curve analysis, we observed that the baseline CD4+ and CD8+ T-cell densities in tumours had higher AUCs (0.73 and 0.71, respectively) than those of Treg cell density and PD-L1 expression (0.67 and 0.60, respectively), indicating their potential as predictive biomarkers for combination therapy with atezolizumab plus carboplatin and nab-paclitaxel (Figure 4).

## 3. Discussion

In this study, we established a QSP model for NSCLC based on the published IO platform and validated it using the data from the IMpower131 clinical trial and alternative dosing regimens of atezolizumab. We explored the biological mechanisms of a combined therapy constituting carboplatin, nab-paclitaxel, and atezolizumab on tumour dynamics. Carboplatin and nab-paclitaxel induce immunogenic cell death, triggering and enhancing the anti-tumour immune response of atezolizumab, which can help rejuvenate T-cells and rescue them from exhaustion, reinvigorating their response against cancer cells. Our established QSP model integrated nonclinical and clinical data, allowing for the identification of potential predictive biomarkers and prediction of the efficacy of treatment of NSCLC.

The model structure was adapted from a previously published IO platform with the necessary modifications. Compared with previous QSP models of NSCLC [7,15], the primary advantage of our model was its ability to predict the efficacy of combination therapies involving checkpoint inhibitors and two chemotherapeutic agents, that is, the integration of atezolizumab plus carboplatin and nab-paclitaxel modules. The mechanisms of action of carboplatin and nab-paclitaxel, including their cytotoxic and antiangiogenic activities, were included in the QSP model, allowing for the estimation of the overall effect of carboplatin and nab-paclitaxel on tumour dynamics.

In this study, we assumed that most of the model parameters for NSCLC and TNBC were similar for the immune system. Nevertheless, certain parameter adjustments were needed [16,17]. For example, disparities in cancer cell dynamics, such as the clinically observed higher cancer cell growth rate and smaller cancer cell diameters in NSCLC than in TNBC, necessitated the re-estimation of our model parameters [18,19].

In the QSP model, the dynamic change in the number of cancer cells was modelled as the combination of the growth rate of cancer cells and killing effect of drugs, including atezolizumab, paclitaxel, and carboplatin (Equation (1)), so that the change in tumour volume could be estimated based on the number of cancer cells (Equation (2)). The effect of various different combination therapy regimens could be further evaluated based on the change in tumour volume according to the Response Evaluation Criteria in Solid Tumours (RECIST) v.1.1 guidelines. The estimation of the cancer cell growth rate in this study was comparable to that in the QSP model of NSCLC developed by Wang et al. [7] (geometric mean ± standard deviation: 0.012 ± 1 versus 0.007 ± 1) after modification. Differences in the characteristics of patient population in each study may lead to variations in the estimated values of cancer cell growth rate. As the cancer cell growth rate is crucial for an increase in tumour volume (Figure 2), meticulous considerations need to be investigated in future research.

Our study verified that three alternative atezolizumab dosage strategies (840 mg Q2W, 1200 mg Q3W, and 1680 mg Q4W) combined with chemotherapy (carboplatin at an AUC of 6 mg/mL/min IV on day 1 plus nab-paclitaxel at 100 mg/m^2^ IV on days 1, 8, and 15) yielded similar ORR. Mechanistically, three types of atezolizumab regimens were found to excessively rescue immune suppression caused by mAPC PD-L1 in tumours and ameliorate the inhibition of the immune response, thereby fully enabling the maturation and activity of Teff cells. Hence, no difference in efficacy was observed among the three regimens of atezolizumab combined with chemotherapy. The alignment of clinical outcomes with the predictions of the QSP model further confirms the reliability of the model with respect to evaluating the efficacy of varying atezolizumab regimens dosages.

Moreover, CD8+ and CD4+ T-cell densities in the tumour were identified as the two most optimised predictive biomarkers, consistent with observations by studies on single-agent PD-L1 blockade therapies [20,21]. This could be attributed to the fact that tumours with low T-cell infiltration density are prone to immune escape, resulting in poor efficacy of ICI monotherapy [22]. A high density of T cells in the tumour reflects relatively high immune recognition of tumour cells in a patient, indicating a T-cell-inflamed tumour microenvironment [23]. Other biomarkers associated with the tumour immune microenvironment and intrinsic features of tumour cells, such as tumour mutational burden and mismatch repair deficiency, are worth further investigation [24].

Our study has certain limitations that warrant further discussion. First, our model simplified its molecular and cellular mechanisms because of the intricate nature of the immune system and limited clinical data. Second, the model simulations may not fully replicate the complexity of a real clinical situation, such as the formation of new metastatic lesions. With further accumulation of safety data and enhanced understanding of the mechanisms underlying adverse events, we can anticipate the development of QSP models dedicated to predicting the toxicity of drug combinations. Such advancements would enable the determination of a more nuanced balance between efficacy and safety, ultimately leading to the optimisation of dosage regimens.

## 4. Materials and Methods

### 4.1. Model Development

#### 4.1.1. Model Structure

The QSP model was built upon previous established models for NSCLC [15] and TNBC [9], and comprised four main compartments: the central compartment, representing the circulation of therapeutic agents and immune cells in the circulating blood; the peripheral compartment, representing peripheral organs/tissues maintaining naïve T cells; the lymph node compartment, representing tumour-draining lymph nodes immediately downstream of the tumour, where T-cell activation occurs; and the tumour compartment, representing the dynamics of cancer cells, activated T cells, APCs, and MDSCs).

Ten modules were built to investigate the dynamics of Treg, helper T cells (Th), APCs, cancer cells, tumour-specific neoantigens and tumour-associated self-antigens, immune checkpoints, CTLA4, MDSCs, nab-paclitaxel, and carboplatin. The structures of all these modules were established according to the QSP model by Wang et al. [9], except for the cancer module, which was modified based on a previous study on NSCLC [15]. Figure 5 shows the interactions among all the compartments and modules. The QSP platform was developed and validated using the SimBiology toolbox in MATLAB (MathWorks, version R2023b, Natick, MA, USA).

#### 4.1.2. Model Parameters

We assumed that all model parameters were consistent between NSCLC and TNBC, except for those pertaining to the dynamics of cancer cells, which required adjustment. The altered parameters were then plugged into the modules of cancer cells, tumour-specific neoantigens, tumour-associated self-antigens, Teff, immunotherapeutic agents, and chemotherapeutic drugs.

Notably, these altered model parameters were assigned based on clinical and experimental data from previous studies, where available (Appendix A). The remaining parameters were calibrated based on PSA and estimated using data from the IMpower131 study.

The IMpower131 study investigated the efficacy and safety of a combination of atezolizumab, carboplatin, and nab-paclitaxel compared to chemotherapy alone (carboplatin and nab-paclitaxel) in the treatment of patients with advanced non-squamous NSCLC. Atezolizumab was administered at 1200 mg IV on day 1, carboplatin at an AUC of 6 mg/mL/min IV on day 1, and nab-paclitaxel at 100 mg/m^2^ IV on days 1, 8, and 15. No drug-drug interaction was observed between nab-paclitaxel and carboplatin [25]. Therefore, we hypothesised that carboplatin would reach a stable concentration in the circulating blood and enter the tumour cells at a constant proportion. The impact of the parameters on tumour volume, a major clinical indicator of disease progression, was determined using PSA. The investigated parameters were randomly and simultaneously generated using Latin Hypercube Sampling (LHS) and then plugged into the model to simulate the tumour volume in patients with NSCLC. The partial rank correlation coefficient was estimated for each parameter.

The most influential parameters and their distribution were further estimated by fitting the ORR and DOR from the IMpower131 study according to the guidelines of the RECIST v.1.1 [26]. To match the clinical trial settings of the IMpower131 study, the treatment duration was set to 400 d and tumour volumes were estimated every 8 weeks.

The reported ORR was 49.7% (95% CI: 44.3–55.1%) for the atezolizumab plus carboplatin and nab-paclitaxel arm and 41.0% (95% CI: 35.7–46.6%) for the carboplatin plus nab-paclitaxel arm, with a median DOR of 7.3 (95% CI: 6.8–9.5) months and 5.2 (95% CI: 4.4–5.6) months, respectively.

### 4.2. Model Evaluation

The LHS method was employed to create 500 virtual patients. ORR and DOR were estimated based on the established model and then compared to those observed in the IMpower131 study.

Due to the flat exposure–response relationship and a wide therapeutic window, three dosing regimens of atezolizumab (840 mg Q2W, 1200 mg Q3W, and 1680 mg Q4W) were approved by FDA. Alternative dosing regimens provide convenience and flexibility to patients and may also reduce the frequency of visits and associated costs, easing patients. Notably, alternative regimens should not result in a significant difference in percentage change in tumour volume [27]. Based on this, the model was further validated using an atezolizumab alternative dosing regimen that consisted of 840 mg Q2W, 1200 mg Q3W, and 1680 mg Q4W on day 1, along with nab-paclitaxel at 100 mg/m^2^ IV on days 1, 8, and 15 and carboplatin at an AUC of 6 mg/mL/min IV on day 1. A virtual cohort of 1000 patients for each dosage regimen was created using the LHS method. The tumour volume and percentage change in tumour diameter from the baseline were estimated for each treatment regimen.

### 4.3. Model Application

Based on the subgroup analysis results of the IMpower131 study [12] and the observations from other clinical studies on single-agent PD-L1 blockade therapies [20,21], we screened six indicators, including PD-L1 expression, CD4+ T-cell density, CD8+ T-cell density, Treg cell density, CD4+/Treg, and CD8+/Treg, to identify potential biomarkers.

The administration of a combination therapy consisting of atezolizumab at 1200 mg intravenously (IV) on day 1, carboplatin at an AUC of 6 mg/mL/min IV on day 1, and nab-paclitaxel at 100 mg/m^2^ IV on days 1, 8, and 15 to 1000 patients was simulated using the established QSP model. These patients were subsequently categorised into response and non-response subgroups in accordance with the RECIST v.1.1 guidelines [26]. The Wilcoxon test was performed to compare the response among different subgroups by the established model.

ROC curve analysis was used to assess the ability of a biomarker to distinguish between response and non-response subgroups. ROC curve analysis was performed using the “calculate_roc” function in MATLAB (MathWorks, version R2023b).

## 5. Conclusions

We have extended the functionality of the existing QSP model to encompass both ICI monotherapy and platinum-containing combined chemotherapy regimens. After calibration and validation, the established QSP model could adequately characterise the biological mechanisms of action of the triple combination of atezolizumab, nab-paclitaxel, and carboplatin in patients with NSCLC, and identify predictive biomarkers for precision dosing. This substantial enhancement widens the applicability of the QSP model and brings it into closer alignment with clinical practice. As a tool for advancing our understanding of the connections between the mechanisms of action of therapeutic regimens and clinical outcomes, this QSP model holds immense value in future cancer research. It has the potential to serve as a fundamental model platform for drug development and personalised treatment strategies, particularly for therapies that integrate single-agent ICIs with platinum-doublet chemotherapy regimens.

## Figures and Tables

**Figure 1 pharmaceuticals-17-00238-f001:**
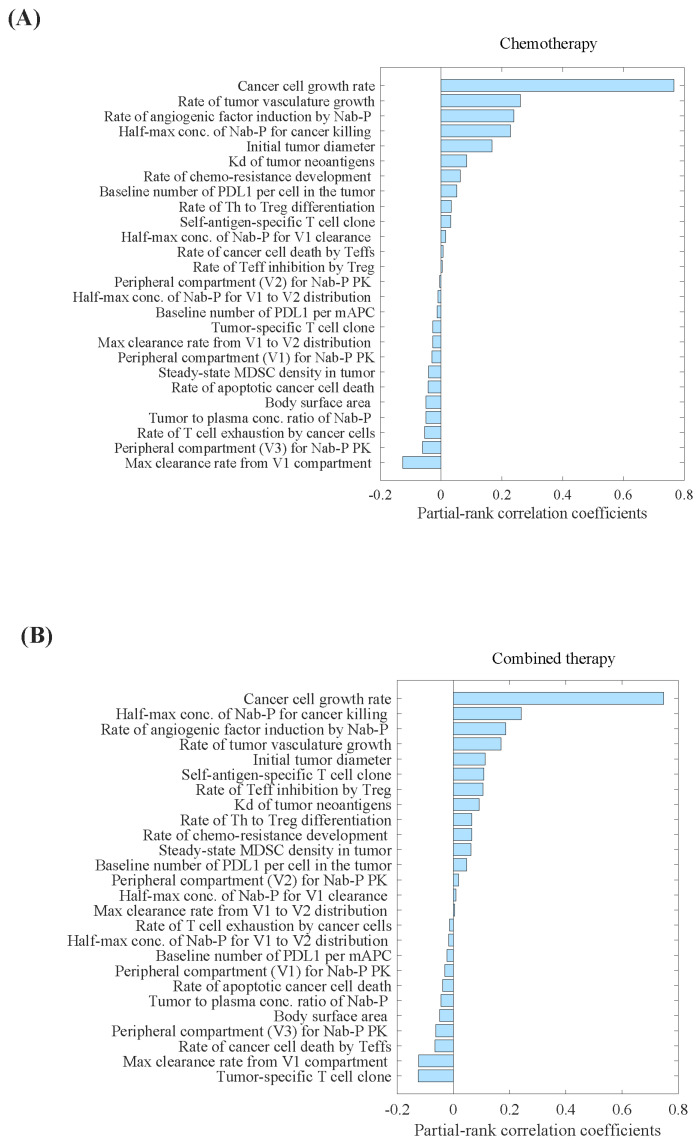
Parameter sensitivity analysis for tumour volume in (**A**) carboplatin plus nab-paclitaxel group, and (**B**) atezolizumab plus carboplatin and nab-paclitaxel group. Parameter sensitivity analysis was performed by varying a set of 26 parameters simultaneously and performing partial correlation analysis to evaluate the effect of those inputs on the model outputs, primarily percentage change in the tumour volume. Kd, binding affinity; mAPC, mature antigen presenting cell; MDSC, myeloid- derived suppressor cells; Nab-P, nab-paclitaxel; Teff, effector T cell; Th, T helper cell; Treg, regulatory T cell; V1, the central compartment; V2, the first peripheral compartment; V3, the second peripheral compartment.

**Figure 2 pharmaceuticals-17-00238-f002:**
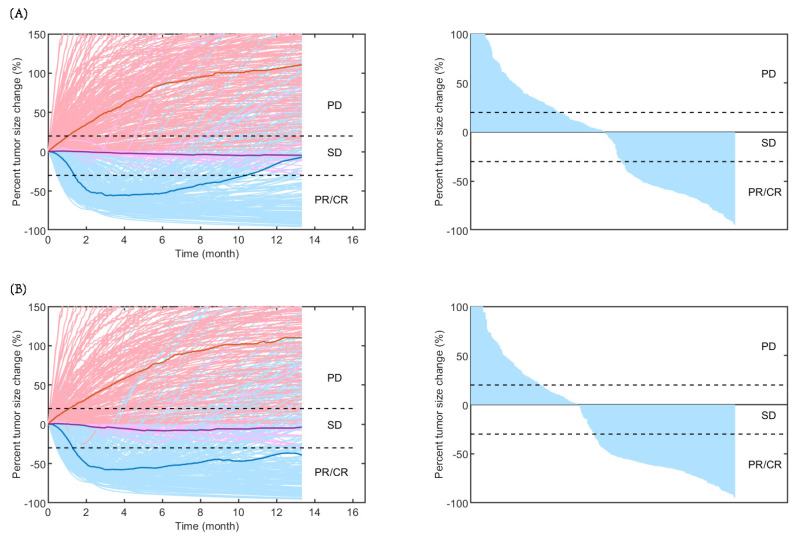
Rate of response (**left**) and the best overall response (**right**) in model-predicted tumour diameter of 500 randomly virtual patients. Response is assessed by RECIST V.1.1 in (**A**) carboplatin plus nab-paclitaxel group and (**B**) atezolizumab plus carboplatin and nab-paclitaxel group. Median (thick lines) and individual (thin line) rate of response are shown in PD (red), SD (purple), and PR/CR (blue) subgroups. CR, complete response; PD, progressive disease; PR, partial response; SD, stable disease.

**Figure 3 pharmaceuticals-17-00238-f003:**
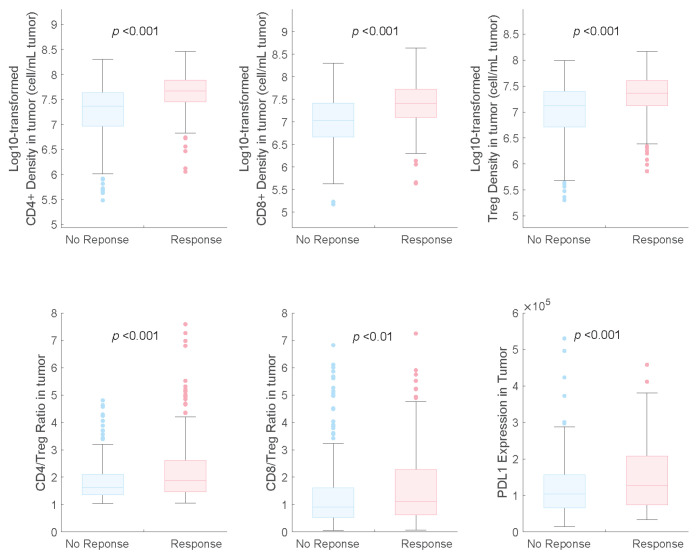
Pretreatment distributions of potential predictive biomarkers in responders and non-responders in combination therapy. NR, non-responders; R, responders.

**Figure 4 pharmaceuticals-17-00238-f004:**
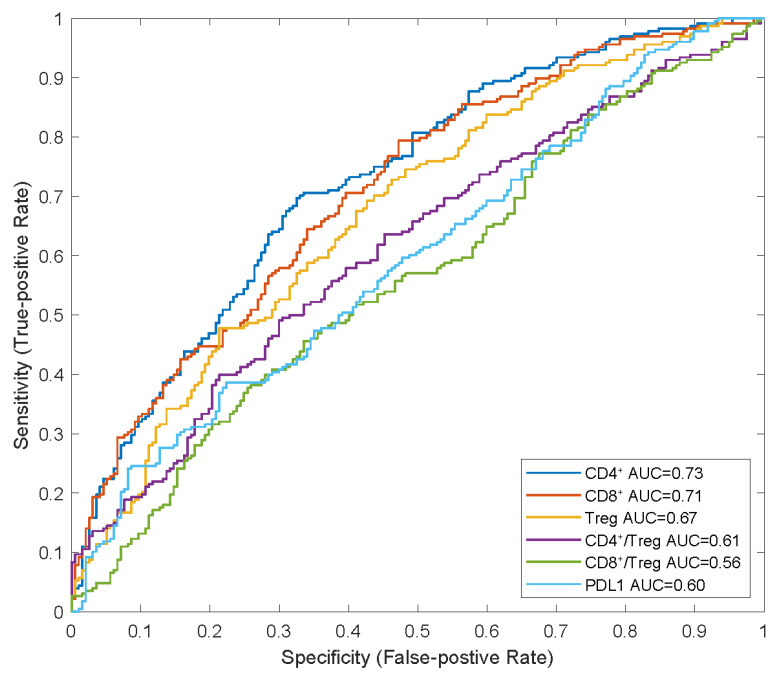
ROC analysis of potential predictive biomarkers in combination therapy. APCs, anti-gen-presenting cells; AUC, areas under curve; NR, non-responders; R, responders; ROC, receiver operating characteristic.

**Figure 5 pharmaceuticals-17-00238-f005:**
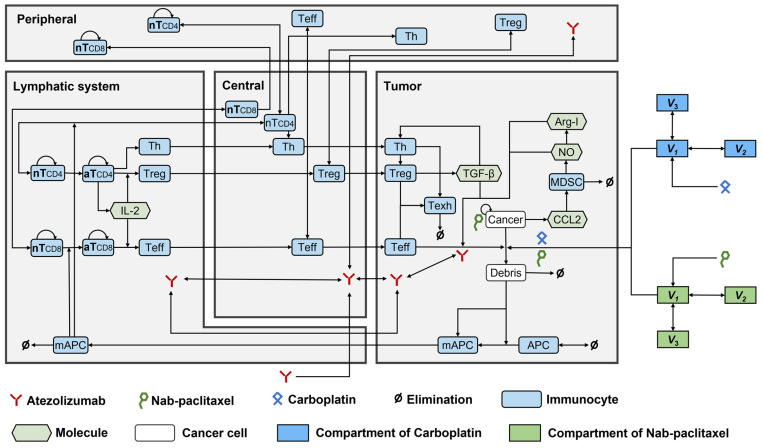
The dynamics of the major species in the quantitative systems pharmacology model of patients with NSCLC. APC, antigen-presenting cell; Arg-I, arginase I; aT, activated T cell; mAPC, mature antigen presenting cell; MDSC, myeloid-derived suppressor cell; NO, nitric oxide; nT, naïve T cell; Teff, effector T cell; Texh, exhausted T cell; Th, T helper cell; Treg, regulatory T cell.

**Table 1 pharmaceuticals-17-00238-t001:** Efficacy prediction for the virtual patient cohort generated based on calibrated parameter distribution.

Therapy Regimen	Carboplatin Plus Nab-Paclitaxel		Atezolizumab Plus Carboplatin and Nab-Paclitaxel	
	Prediction (95% CI)	Observation (95% CI)	Prediction (95% CI)	Observation (95% CI)
ORR, %	40.5 (37.1–43.9)	41.0 (35.7–46.6)	52.0 (45.8–55.4)	49.7 (44.3–55.1)
Median DOR (month)	5.3 (5.3–7.1)	5.2 (4.4–5.6)	7.1 (5.3–8.8)	7.3 (6.8–9.5)

Abbreviations: CI, confidence interval; ORR, objective response rate; DOR, duration of response.

**Table 2 pharmaceuticals-17-00238-t002:** Percentage change in tumour diameter (A) and tumour volume (B) for different dosing regimens of various doses and schedules by model simulation.

Value	Atezolizumab Regimens	Chemotherapy Regimens	Median	25th Percentiles	75th Percentiles
Change in tumour diameter (%)	840 mg Q2W	Nab-paclitaxel 100 mg/m^2^ on days 1, 8, and 15 and carboplatin at an AUC of 6 mg/mL/min on day 1	−10.97	−49.40	20.58
	1200 mg Q3W		−7.33	−47.89	19.86
	1680 mg Q4W		−7.73	−46.92	21.45
Tumour Volume (mm^3^)	840 mg Q2W		6.72	0.68	22.06
	1200 mg Q3W		7.23	0.74	21.93
	1680 mg Q4W		7.14	0.80	21.50

Abbreviations: AUC, area under the concentration–time curve.

## Data Availability

The original data that support the findings of this study can be available from the corresponding authors upon reasonable request, without undue reservation.

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
