# Peer review of "Development and Evaluation of a Quantitative Systems Pharmacology Model for Mechanism Interpretation and Efficacy Prediction of Atezolizumab in Combination with Carboplatin and Nab-Paclitaxel in Patients with Non-Small-Cell Lung Cancer"

_pharmaceuticals, 2024, doi:10.3390/ph17020238_

Round 1

Reviewer 1 Report

Comments and Suggestions for Authors

The manuscript presented a combination therapy as lung cancer treatment. I have few comments to make.

In Figure 2, rate of tumor growth is shown as parameter against tumor volume. One would expect that with increase in rate volume would increase proportionally and hence a biased variable. What is the significance of this variable?

Was  atezolizumab dosages vary while the carboplatin and nab-paclitaxol doses were kept fixed? Or all of them were varied and combined does-responses were incorporated in the model?

Increased atezolizumab dosages have same effect on tumor volume as per the authors. What is the significance of this finding? How does it connect to the mechanism of combination therapy model reported by the authors?   

Author Response

The manuscript presented a combination therapy as lung cancer treatment. I have few comments to make.

In Figure 2, rate of tumor growth is shown as parameter against tumor volume. One would expect that with increase in rate volume would increase proportionally and hence a biased variable. What is the significance of this variable?

Response: Thank you for your advice. Rate of tumor growth is misleading. We replaced rate of tumor growth with cancer cell growth rate and revised the manuscript as below:

Line 294 (4. Discussion)

In the QSP model, the dynamic change in the number of cancer cells was modelled as the combination of the growth rate of cancer cells and killing effect of drugs, including atezolizumab, paclitaxel, and carboplatin (Eq. 1), so that the change in tumour volume could be estimated based on the number of cancer cells (Eq. 2). The effect of various different combination therapy regimens could be further evaluated based on the change in tumour volume according to the RECIST v.1.1 guidelines. The estimation of the cancer cell growth rate in this study was comparable to that in the QSP model of NSCLC developed by Wang et al.[7] (geometric mean ± standard deviation: 0.012 ± 1 versus 0.007 ± 1) after modification. Differences in the characteristics of patient population in each study may lead to variations in the estimated values of cancer cell growth rate. As the cancer cell growth rate is crucial for the increase in tumour volume (Figure 2), meticulous considerations need to be investigated in future research.

Was atezolizumab dosages vary while the carboplatin and nab-paclitaxol doses were kept fixed? Or all of them were varied and combined does-responses were incorporated in the model?

Response: We appreciate your valuable comments and clarified this issue in the manuscript:

Line 135 (2.2. Model evaluation)

…Based on this, the model was further validated using an atezolizumab alternative dosing regimen that consisted of 840 mg Q2W, 1200 mg Q3W, and 1680 mg Q4W on day 1, along with nab-paclitaxel at 100 mg/m2 IV on days 1, 8, and 15 and carboplatin at an AUC of 6 mg/mL/min IV on day 1. A virtual cohort of 1000 patients for each dosage regimen was created using the LHS method.

Line 146 (2.3. Model application)

The administration of a combination therapy consisting of atezolizumab at 1,200 mg intravenously (IV) on day 1, carboplatin at an AUC of 6 mg/mL/min IV on day 1, and nab-paclitaxel at 100 mg/m2 IV on days 1, 8, and 15 to 1000 patients was simulated using the established QSP model.

Line 242 (3.2. Model evaluation)

Moreover, three atezolizumab dosage regimens (840 mg atezolizumab Q2W, 1200 mg atezolizumab Q3W, and 1680 mg atezolizumab Q4W) combined with the same chemotherapy regimens (carboplatin at an AUC of 6 mg/mL/min IV on day 1 and nab-paclitaxel at 100 mg/m2 IV on days 1, 8, and 15) led to similar changes in tumour volume and tumour diameter (Table 2).

Table 2. Percentage change of tumour diameter (A) and tumour volume (B) for different dosing regimens of various doses and schedules by model simulation.

Value

Atezolizumab regimens

Chemotherapy regimens

Mean

Median

25th Percentiles

75th Percentiles

Change of tumour diameter (%)

840 mg Q2W

Nab-paclitaxel 100 mg/m2 on days 1, 8, and 15 and carboplatin at an AUC of 6 mg/mL/min on day 1

-5.91

-10.97

-49.40

20.58

1200 mg Q3W

-3.42

-7.33

-47.89

19.86

1680 mg Q4W

-3.89

-7.73

-46.92

21.45

Tumor Volume (mm3)

840 mg Q2W

16.18

6.72

0.68

22.06

1200 mg Q3W

16.55

7.23

0.74

21.93

1680 mg Q4W

17.31

7.14

0.80

21.50

Abbreviations: AUC, area under the concentration-time curve.

Increased atezolizumab dosages have same effect on tumor volume as per the authors. What is the significance of this finding? How does it connect to the mechanism of combination therapy model reported by the authors?

Response: Thank you for your comments.

Line 129 (2.2. Model evaluation)

Due to the flat exposure-response relationship and a wide therapeutic window, three dosing regimens of atezolizumab [840 mg every 2 weeks (Q2W), 1200 mg every 3 weeks (Q3W), and 1680 mg every 4 weeks (Q4W)] were approved by FDA. Alternative dosing regimens provide convenience and flexibility to patients and may also reduce the frequency of visits and associated costs, easing patients. Notably, alternative regimens should not result in a significant difference in percentage change in tumour volume [20]…

Reference

  1. LIU, W.; XUE, J.; YU, Z.; WANG, Z.; CHEN, R.; ZHOU, T. Interpretation of pharmacokinetic-based criteria for supporting alternative dosing regimens of programmed cell death receptor-1 (PD-1) or programmed cell death-ligand 1 (PD-L1) blocking antibodies for treatment of patients with cancer guidance for industry. Chinese Journal of Clinical Pharmacology and Therapeutics 2022, 27, 86.

Line 309 (4. Discussion)

…Mechanistically, three types of atezolizumab regimens were found to excessively rescue immune suppression caused by mAPC PD-L1 in tumours and ameliorate the inhibition of the immune response, thereby fully enabling the maturation and activity of Teff cells. Hence, no difference in efficacy was observed among the three regimens of atezolizumab combined with chemotherapy. The alignment of clinical outcomes with the predictions of the QSP model further confirms the reliability of the model with respect to evaluating the efficacy of varying atezolizumab regimens dosages.

Reviewer 2 Report

Comments and Suggestions for Authors

The manuscript was aimed to illustrate a quantitative systems pharmacology  model to predict an efficacy ofcombined triple-therapy for non-small cell lung cancer (NSCLC). The triple therapy was composed of single immune checkpoint inhibitor (ICI) atezolizumab supplemented with 2 chemotherapeutic agents (carboplatin and nab-paclitaxel). 

In general, the manuscript is well-prepared. However, the major concern  about this manuscript is the absence of the overall algorithm illustrating the relationship between the immune profile of tumors and benefits of the triple-therapy analysed in this manuscript. Indeed, the authors are describing the obvious findings illustrating that CD4+ and CD8+ T cell counts and Tregs cell densities in tumors are significantly higher in responders that in non-responders. Similar findings were observed for PDL-1 expression. However, all these markers are well-known as the beneficial predictive markers for immunotherapy of cancers even used alone (without chemotherapy). The authors have to clear the novelty of their findings and highlight them in the abstract and conclusions, as well.   

Author Response

The manuscript was aimed to illustrate a quantitative systems pharmacology model to predict an efficacy of combined triple-therapy for non-small cell lung cancer (NSCLC). The triple therapy was composed of single immune checkpoint inhibitor (ICI) atezolizumab supplemented with 2 chemotherapeutic agents (carboplatin and nab-paclitaxel). In general, the manuscript is well-prepared.

Response: We appreciate the reviewer’s recognition of our works.

However, the major concern about this manuscript is the absence of the overall algorithm illustrating the relationship between the immune profile of tumors and benefits of the triple-therapy analyzed in this manuscript.

Response: Thank you for your advice.

Line 162 (3.1.1. Model structure)

The primary compartments and modules of the QSP model were constructed based on prior TNBC studies. Within the QSP model, the dynamics of cancer cell growth and death was described by the cancer module. The antigen module described the release of both self and cancer-associated antigens from dead cancer cells, whereas the APC module described the antigens captured by mature antigen-presenting cells (mAPC) migrating via the lymphatic vessels to the tumour-draining lymph nodes (TdLNs). The Teff and Treg module described the mAPC-mediated activation of both naïve CD8+ and CD4+ T-cells, converting them into Teffs and Tregs, respectively. Following activation, Teff and Treg cells are known to leave the lymph nodes, circulate through the bloodstream, and, upon extravasation, localize to the tumour and other peripheral tissues. Within the tumour microenvironment, Teff cells engage in the direct elimination of cancer cells but can become exhausted due to suppressive actions by Tregs and the PD-1/PD-L1 interaction with tumour cells. However, atezolizumab enhances the anti-tumour response by preventing the PD-1/PD-L1 interaction, thereby facilitating the unrestricted activity of Teff cells against tumour cells, as described by the checkpoint module. Additionally, the nab-paclitaxel and carboplatin modules described the targeting of cancer cells via the cytotoxic effects of nab-paclitaxel and carboplatin. The established QSP model comprised 216 ordinary differential and 55 algebraic equations. These model equations are presented in Supplementary Tables S2-S7.

Indeed, the authors are describing the obvious findings illustrating that CD4+ and CD8+ T cell counts and Tregs cell densities in tumors are significantly higher in responders that in non-responders. Similar findings were observed for PDL-1 expression. However, all these markers are well-known as the beneficial predictive markers for immunotherapy of cancers even used alone (without chemotherapy). The authors have to clear the novelty of their findings and highlight them in the abstract and conclusions, as well.

Response: Thank you for your advice.

Line 28 (Abstract)

…This significant extension of the QSP model not only broadens its applicability but also more accurately reflects real-world clinical settings. Importantly, it positions the model as a critical foundation for model-informed drug development and the customization of treatment plans, especially in the context of combining single-agent ICIs with platinum-doublet chemotherapy.

Line 334 (5. Conclusions)

We have extended the functionality of the existing QSP model to encompass both ICI monotherapy and platinum-containing combined chemotherapy regimens. After calibration and validation, the established QSP model could adequately characterize the biological mechanisms of action of the triple combination of atezolizumab, nab-paclitaxel, and carboplatin in patients with NSCLC, and identify predictive biomarkers for precise dosing. This substantial enhancement widens the applicability of the QSP model and brings it into closer alignment with clinical practice. As a tool for advancing our understanding of the connections between the mechanisms of action of therapeutic regimens and clinical outcomes, this QSP model holds immense value in future cancer research. It has the potential to serve as a fundamental model platform for drug development and personalized treatment strategies, particularly for therapies that integrate single-agent ICIs with platinum-doublet chemotherapy regimens.
